# Crop Insurance, a Frugal Innovation in Tanzania, Helps Small Maize Farmers and Contributes to an Emerging Land Market

## Meine Pieter van Dijk

Maastricht School of Management, 6229 EP Maastricht, The Netherlands; dijkm@msm.nl; Tel.: +31-433870808

**Abstract:** A land market is emerging in Tanzania, triggered by initiatives to reform land legislation and modernize agriculture through frugal innovations, combining hybrid seeds and weather-based index insurance with the use of mobile telephones. The analysis shows that agricultural modernization can be a driver for an emerging land market. Demand for land increases and because of the liberalization of land rights, land can be bought or leased, something the more successful farmers do. To assess the effects of crop insurance for maize farmers, a frugal innovation, a survey has been carried out in three regions. Two hundred farmers were interviewed using cluster sampling with the villages as sampling units and then selecting households per village. The rural transformation process, driven by innovation, started with the development of an ecosystem and land registration while allowing more private (commercial and non-commercial) initiatives. The triggers are frugal innovations. Crop insurance, combining existing hybrid seeds, with satellite images and mobile telephones, brings about a transformation process and pumps money into the land system. People noticing that hybrid maize works, if you have hybrid seeds, the complementary inputs, and an insurance policy, jump on the band wagon, which leads to more demand for land and contributes to an emerging land market.

**Keywords:** emerging land market; land policies; ecosystems; agricultural innovation; frugal innovation; hybrid maize; crop insurance; Tanzania

## 1. Introduction

Climate change and rapid population growth force African countries to reconsider their agricultural and land policies. Some countries go for precision agriculture to increase agricultural production. In Rwanda the use of the exact quantity of fertilizers, pesticides, and insecticides (preferably bio-based) is promoted, which eventually means the farmers may need less land [1]. Agricultural innovations can lead to different and more intensive land use, in particular when green houses are introduced.

In this study, we analyze the effects of new legislation for land transactions, the effects of a developing ecosystem, and of frugal innovation for small maize farmers in Tanzania on the functioning of the rural land market [2]. Existing innovations, such as the use of hybrid seeds, satellite weather images, and mobile telephones, were combined when introducing crop insurance for maize farmers, which would boost investments by the farmers [3].

New technological options were introduced in Tanzania through a number of projects. To determine their effects on land markets, we studied the increased demand for land through the introduction of crop insurance for maize farmers. The changes were the effects of increased activities of private (commercial and non-commercial) actors, wanting to insure as many farmers as possible at minimum cost against possible droughts. The crop insurance project concerned a frugal innovation because it combined the use of available hybrid maize seed, with a loan and technical assistance and used a Weather Index Insurance (WII) based on available satellite images to determine whether rains were insufficient in the areas concerned during the agricultural cycle (planting, germination, and ripening). If less rain than normal is observed, farmers registered through their mobile phones will be

compensated for the loss of money spent on hybrid seeds and fertilizers. This can happen by topping up the credit on their mobile phones, which they can use for money transfers, thanks to MPesa-type payment systems [4]. This is a typical frugal innovation, using a combination of existing technologies in an intelligent way [5]. This was possible because most farmers in Africa now have mobile phones. They can insure a bag of hybrid seeds bought from Seedco (the seed company) by sending an SMS message. It only covers the germination period (the first 28 days) in this case. Alternatively, they can sign up for a package of inputs for one acre of land provided through a nongovernmental organization (NGO; 1AF), or they can use a commercial intermediary to get a crop insurance policy.

These are the three modalities we studied at the request of SBCF (the Swiss Capacity Building Facility), an NGO financed by Swiss insurance companies. Training projects were initiated by SCBF, through an NGO, Acre Africa AA [6], with the objective to introduce the crop insurance idea in Tanzania. This international NGO AA had a local affiliate (the One Acre Fund, 1AF, Tanzania) undertaking the actual training of their staff and eventually the farmers. Thousands of farmers were trained in the three regions studied. The training concerned how to deal with hybrid seeds and how to handle the risk of no rainfall. Drought means the loss of the money spent on seed, fertilizers, and labor. More than 20,000 farmers were insured in the Iringa region and more than 10,000 in total in the other regions studied, Mwanza and Arusha. The insurance is a weather-based insurance, covering the risk of not having enough rainfall. Between 2011 and 2014, SBCF funded four projects in Tanzania introducing crop insurance for maize farmers in the Mwanza, Iringa, and Arusha regions. Training was provided by local organizations and the introduction of crop insurance was usually part of a package of frugal innovations offered to the maize farmers, male or female.

Tanzania had a 60 million population in 2020. However, there is no acute scarcity of land, rather water is becoming a bottleneck in certain areas. The lack of productivity of many small farmers is the main underlying problem in Tanzania, which impacts both household income and the food security of the country. Agricultural development is stimulated by the government and helped by export opportunities to countries of the East African Community, to which Tanzania belongs. Fifty-four percent of Tanzania's land is classified as agriculture of which 14.3% is arable, 2.3% is covered with permanent crops, and 27.1% is permanent pasture [7].

## 2. Relevant Theories

Land markets in emerging economies are often depicted as functioning in a suboptimal way [8]. According to Mugabi [9], Tanzania faces a number of challenges. The literature points out that land security would promote investment and land policies are part of the ecosystem. The ecosystem could be one of the drivers of change in the land market, just like the innovations generated and distributed through this system. Adner defined an ecosystem as "the alignment structure of the multilateral set of partners that need to interact in order for a value proposition to materialize" [10].

The following challenges concerning land ownership in Tanzania are mentioned in the literature: farmer–pastoralist conflicts [11], tenure disputes [12], alienation of peasants [13], and land-use conflicts [14]. Bergius et al. described such conflicts in the Southern Agricultural Growth Corridor of Tanzania (SAGCOT), where the government with the aid of donors and investors tried to implement the green economy in Africa. The area is dominated by investors who combine commercial agriculture with environmental conservation [14]. The authors concluded that "some smallholder farmers are deposed through these expansions, others are contracted as outgrowers", but in particular the pastoralists are suffering.

In 1999, Tanzania passed a series of land laws and regulations that granted customary rights of occupancy (CCROs) as an equal status to other property rights, or de facto ownership of land. Biddulph and Hillbom [15] summarized it as: "the law offers registration of private interests in the form of certificates of customary rights of occupancy". These CCROs fit within a broader community land approach in Tanzania. We will study what

this means for the functioning of the land market in the rural areas and the issue also dealt with by Yefred [16]. The security of land tenure for holders of the right of occupancy in Tanzania under the Land Policy 1995 and the Land Act of 1999 is assured and this should stimulate investments and functioning land markets.

Tanzania's land policy and land reform have also been studied by Hayuma and Conning [17], who were looking at the positive effects on competitiveness in the private sector of Tanzania's land policy and land reform. Kabigi et al. [18] emphasized the resulting land security of the reforms and the widening of the range of rights once a CCRO has been obtained. Questions asked were: what about land disputes, are farmers able to sell the land, and can land be used as security for an agricultural loan? They claimed that neo-cadasters partly replace customary land institutions but are not yet effective or sustainable. We start at the other end, by asking the farmers whether they buy or lease land, which will provide data on whether a land market is developing. If they declare in a significant number of cases that land is bought and sold and that they can buy or lease it, we take it as an indication of an emerging land market. Wang [19] used this term in the Chinese context for increased land commercialization after marketization and other land policy reforms.

In this article, we assess the effects of changes in legislation, a developing ecosystem, and the introduction of frugal innovations in Tanzania's agriculture on the functioning of the rural land market. Did it become a market where dynamic farmers can buy or lease land to work on? Do farmers sell land or lease it out if they do not need it or have too much land for the number of household members that can work on it? We will study the interaction between the innovation of providing improved seeds, other inputs, technical advice, and crop insurance and the functioning of the local land market. In the first stage of modernization of agriculture, farmers want more land, and this may become available if proper land policies are in place.

Kariuki et al. [11] analyzed changes in land use in East Africa, identifying several drivers of change in land use. However, they did not point to agricultural innovation as a driver. We will study whether providing a frugal innovation can be considered the driver for a change in the functioning of the land market.

Droughts occur more frequently in Tanzania due to climate change. They contribute to farmers' risks, but their main problem is low agricultural productivity. Extension services are not functioning properly [20]. Most farmers are still using traditional seeds instead of higher-yielding varieties, which could contribute to improving food security in the country. Hence, introducing hybrid maize seed was very important. This is what this combined commercial and non-commercial private sector initiative is doing. It helps Tanzanian farmers by providing crop insurance, which is part of the climate smart approach (CSA) to agricultural development. The expectation is that tenure security leads to more investments and the frugal innovations adoption functions as a driver for a more dynamic land market.

## 3. Materials and Methods

Our overall research questions were: what is the best way to help traditional small maize farmers to increase maize production and what would be the role of land in this process? To assess the effects of a number of frugal innovations on Tanzanian maize farmers a survey was undertaken by the author in the Arusha, Mwanza, and Iringa regions. In total, 200 farmers were interviewed, using a precoded questionnaire with some open questions. The sample relates to the 30,000 farmers receiving crop insurance (0.66%). We did cluster sampling with the villages as sampling units. Subsequently, heads of farmers' households were selected per village as randomly as possible. The objective was to analyze the effects of the innovation's introduced with the support of SCBF and in particular its effects on agricultural productivity and on the income and assets of the concerned households, including the quantity of land used for agriculture.

Land is an important asset for these farmers and our study analyses the role of land, owned or leased, and the prices paid for the land. In addition, the role of the size of the holdings was analyzed to find out whether there is something like a functioning

land market in Tanzania and whether bigger farmers benefit relatively more from the opportunity to acquire hybrid seeds and complementary inputs and to insure themselves for crop failure. The research also identified the importance of land policies for the success of the introduction of crop insurance for maize farmers in Tanzania and looked at the emerging ecosystem.

The following detailed research questions were formulated:

1. What are the effects of current land policies and land reforms in Tanzania?
2. Is there an ecosystem developing, including an emerging land market, to serve small farmers in Tanzania?
3. What is the overall effect of the land policies, the emerging ecosystem, and the promotion of frugal innovations on the functioning of land markets in Tanzania?

## 4. Results

### 4.1. Farmers' Profile

Most farmers in our sample were not just maize farmers but engaged in a number of activities, ranging from growing other cash crops to keeping animals or running a small business. On average, the 200 farmers interviewed were involved in 2.8 different economic activities. The focus of the interviews was on maize production, but we found diversification was also a strategy to mitigate risks related to climate change. If maize prices are low, other agricultural and non-agricultural activities help the farmers to survive and feed their families. When maize prices increase, as in the first year of our study, people can earn a significant money using hybrid seeds in combination with crop insurance. Bigger harvests allow them 'to prepare for the next season' (buying the necessary inputs), to pay school fees, and build a house or part of it.

Other agricultural activities than growing maize were growing a real cash crop (cotton or groundnuts), growing vegetables, and keeping cows, pigs, or chickens. One hundred and seven out of two hundred farmers declared such a source of additional income and provided the production and income figures. They earned as much from these secondary activities as from growing maize. Another 29 declared another activity but could not give an estimate of the net returns per month of, for example, running a shop or a local beer brewery. However, this secondary income helped their family to survive a bad harvest.

One hundred and forty farmers claimed there was often not enough rain. The current water supply is about 75% of what they need for their maize.

Irrigation infrastructure is limited in Tanzania, while in many locations there is a need for additional water supply. One hundred and seventy-seven farmers depended completely on the rain. The other 23 farmers could obtain water from a river or lake, but usually depended on a furrow or they brought the water in buckets. The lack of water was a recurrent problem in the discussion with the farmers.

Who is the average farmer? In our sample he may be she (40% of the 200 farmers interviewed were women), having an average age of 47.6 years and a family of in total 6.6 people. Thirteen farmers had not been at school at all, while 84% went to primary school and another 18 had secondary education, while two had been to university. Eighteen farmers (9%) had no mobile phone at all. The phone was stolen, it had broken down or they had no cash to buy one. One hundred and sixty-three (82%) had a simple mobile phone, while the remaining farmers used smartphones (7%), even with the internet (another 3%, or in total 10% of the farmers had smartphones). They often used several providers. Farmers in Iringa, who we asked whether they used the phone for payments, almost always answered yes.

Most farmers hired additional workers to prepare the land, or for sowing, weeding or harvesting. At the average they employed 3.2 men and 1.9 women, paying about 3000 TSh. per day. Poor farmers depended on their family labor, on 'ujama' (traditional solidarity groups), or new groups set up, for example, by the NGO 1AF. These groups helped their members, without payment. The main expenditure was for hybrid seeds. In the case of using SeedCo, this cost 11,000 TSh. for an acre, or in the other modalities, the farmers paid

for the package of fertilizer, hybrid seeds, technical advice, and insurance. The NGO 1AF charges 235,000 TSh. (about 100 Euros) for a package for one acre in the form of a loan to be repaid in a period of ten months. One hundred and eighty-six farmers had an average turnover of 1,060,000 TSh. in the first year (The turnover of the farmers in year one was five times zero because these farmers did not grow maize in that year. In 9 cases we could not get an answer.), while the 197 providing information had an average turnover of 1,079,000 TSh. in the second year, which was slightly more than in the first year, despite the poor weather conditions and the lower average price paid for maize in the second year (The turnover of farming in the second year was only two times zero because these farmers claimed not to have harvested any maize because of the drought. In one case we could not get an answer.).

### 4.2. Land Ownership: An Emerging Land Market?

Farmers interviewed owned an average 4.98 acres during the first year and 5.22 acres during the second year (Table 1). The table also provides information on the number of farmers buying or leasing land and the quantity of land bought or leased in years 1 and 2.

**Table 1.** Increase in land owned and land leased between 2015/2016 and 2016/2017 in acres.

| Year and Change between the 2 Years | Land Owned (Acres) | Land Corrected (Only Owners) | Leased Land (Acres and Number of Farmers) | Leased Land Only (Acres) | Number of Farmers Leasing |
|---|---|---|---|---|---|
| 2015/2016 | 4.98 | 5.79 for 172 of the 200 who owned land | 6.78 (41) | 5.79 (28) | 41, of which 28 do not own land, while 13 added |
| 2016/2017 | 5.22 | 6.03 for 173 of the 200 who owned land | 10.39 (46) | 6.03 (27) | 46, of which 27 owned no land, while 19 added |
| Change | +0.24 | +0.24 one more farmer owns land | +3.61 | +0.24 | 5 more farmers lease land |

The table shows that in a number of cases farmers had bought or leased additional land. In the first year, 28 had no land but leased and in the second year, 27 had no land but leased it. In the second year, the total number of farmers leasing went from 41 to 46, of which 28 (27 in the second year) did not own land themselves and 13 added leased land to the land they owned already in the first year and 19 did so in the second year.

In the first year, the average size of leased land (for 41 farmers) was 6.78 acres, and in the second year (for 46 farmers), 10.39 acres. If corrected for those who also own land, farmers not owning land would have 5.79 acres on average in the first year and 6.03 in the second year. There were around 30 landowners owning more than 10 acres, which is big in this context. In many locations, there was also a need for an additional water supply.

We noted a remarkable increase in the number of farmers leasing and in the amount of land leased. The conclusion is that the farms are relatively small, but more land is available for buying or leasing. Some farmers went into hybrid seeds in a big way, getting additional land as well, but they often suffered in the second year because of less rain and lower prices paid for maize.

### 4.3. The Role of the Ecosystem

Hayumaand Conning [17] studied Tanzania's land policy and land reform as a component of private sector competitiveness. We concluded that the reforms they describe created the conditions for the development of a land market. The hybrid seeds require farmers to buy seeds (instead of using maize from last year's harvest). They also need to buy fertilizers and sometimes pesticides and insecticides. The upcoming ecosystem, supplying these

goods and services, can be described by the three modalities used to introduce the frugal innovations studied and is the result of agricultural modernization in the last decade.

Depending on the region farmers can access insurance by registering for a replanting guarantee (RPG by Seedco) or go for crop insurance for one or more acres of land with or without a package of inputs. The analysis of interviews revealed large differences between the three modalities for introducing crop insurances that we studied:

1. Farmers sign up for a package of inputs for one acre of land through a local NGO (1AF), which provides hybrid seeds, fertilizer, technical advice, and get insurance via the private sector, this is the case in the Iringa region
2. The seed company provides a cover for the first 28 days. This is what SeedCo offers in the Mwanza region, but it turned out not to be sustainable because they withdrew it later on
3. An insurance company is selling crop insurance policies directly. This is the case of Mviwata (the intermediary founded by farmers to create a farmer-to-farmer exchange forum) in the Arusha region, where the farmers complained about a lack of information and a lack of follow-up, once they had taken the insurance policy.

The three ways of supplying insurance are compared in Table 2. Farmers supported by a local NGO; the One Acre Fund (1AF) showed the best results. In this case, the innovations are embedded in an institutional support structure. That structure is non-commercial and close to the farmers. Supply of the germination cover is provided by the seed company Seedco, a profit-oriented intermediary, which gave up providing the insurance after a few years. Finally, we studied a combination of a commercial (insurance company) and a non-commercial organization (a local NGO). All three approaches insured the final risks with UAP (a local commercial insurance company) and worked with a Swiss re-insurance company. The differences between the three modalities are summarized in Table 2.

**Table 2.** Differences between the three modalities in the three regions studied.

| | 1AF in Iringa Region | Mviwata in Arusha | SeedCo in Mwanza |
|---|---|---|---|
| Household size (number of persons) | 5.3 | 11.0 | 6.3 |
| Own land used (acres) | 2.54 | 6.46 | 8.87 |
| Inputs used year 2 ** (TZS) | 376,870 | 179,850 | 337,940 |
| Maize produced (kg) | 2023 | 2996 | 2183 |
| Average yields * (kg/acre) | 1592 | 873 | 492 |
| Farmers interviewed (number of persons) | 108 | 39 | 53 |
| Percentage of women interviewed | 55% | 10% | 32% |
| Average insurance pay-out (TZS) | 20,000 | 32,000 | Replacement of the seeds that failed to germinate |

* Assuming 50% of the land is used for maize. ** Mainly seeds and fertilizers.

The results showed that farmers in the Arusha and Mwanza region had more land, bigger families but lower average yields than farmers in the Iringa region. In the Mwanza region, the farmers had comparatively, the lowest production per acre. In the Arusha region, farmers comparatively spend the lowest amount on inputs used. In contrast, farmers in the Iringa region spend more on inputs and, with the support received from 1AF, they comparatively, get the highest average yields per acre while they cultivate less land than

in other regions. The data showed that the delivery of an all-inclusive package to farmers with regular interaction within an institutional support structure delivers the best result in terms of productivity (average yields), investment (in inputs, land, and equipment), and expressed appreciation of the organizations involved.

Ecosystem development should always be farmer-driven. Larger farmers can afford to obtain the inputs and services from other parts of the country, but for the smaller ones, it needs to be in the region. Partially, we saw the ecosystem developing spontaneously, but often government initiatives or donor projects help to get the institutions in place. NGOs or academic institutions may lobby for it and show that there is a business model for the small farmer if he or she can be convinced to use more and different inputs and take crop insurance. Our study also made an inventory of complaints of farmers, which should be taken seriously. The challenge with the introduction of new technology is to come to scale with the technology and reach the break-even point for the introduction of crop insurance while stimulating the development of a land market. This is happening, but land cannot yet be used as a guarantee for a crop-related loan. There are Agricultural Development Banks, some also government-owned, who can help to reach these farmers with agricultural loans if the legal system promoted taking loans with land as a guarantee. They also could consider new forms of guarantee [4]. They can and will step in, as long as the sales of the farmers are secured and farmers are organized in private groups, for example, by NGOs. The agricultural marketing co-operatives (AMCOS) try to play this role (The agricultural marketing co-operatives (AMCOS) in Tanzania unite smallholder farmers to help them access inputs and other services including storage, processing and marketing of their produce to improve their lives through increased income.). This implies clustering with the support of different actors in the ecosystem: the seed suppliers, the AMCOS, the crop buyers, 1AF, etc. That may also make it easier to access funds and to use land in an optimal way.

### 4.4. Effects of the Frugal Innovation

Does the intervention of introducing new technologies such as hybrid seed with a crop insurance contribute to the development of a land market and increase the income of the farmers? The effect of the introduced crop insurance was analyzed by looking at a number of agricultural variables (Table 3) and some indicators of change in household assets (Table 4).

There were a number of clear improvements between year 1 and year 2 that could partially be attributed to the availability of hybrid seeds and crop insurance. Positive effects were found when comparing data for the first and the second year. For example, farmers used more land and inputs and produced more maize, despite poorer rains and lower maize prices in the second year, as shown in Table 3.

**Table 3.** Differences between agricultural variables in years 1 and 2.

| Variable/Average | First Year | Second Year | Increase in 2 Years |
|---|---|---|---|
| Land used (in acres) | 4.98 | 5.22 | +0.24 |
| Inputs (TZS) | 283,825 | 327,450 | +43,625 |
| Weekly consumption (TZS) | 20,911 | 23,576 | +2665 |
| Monthly farm income (TZS) | 77,665 | 92,161 | +14,496 |
| Production maize (kg) | 1922.76 | 2252.9 | +330.14 |
| Productivity * (kg/Acre) | 772.2 | 863.2 | +91.0 |
| Production turnover (TZS) | 1,060,000 | 1,079,000 | +19,000 |

* Assuming 50% of the land is used for maize.

**Table 4.** Effects of the innovations measured through the value of assets between year 1 and 2.

| Variable/Average | First Year | Second Year | Increase in 2 Years |
|---|---|---|---|
| Investments outside agriculture | 52 out of 121 farmers | 69 of 113 farmers | an increase in farmers investing outside agriculture |
| Value house (TZS) | 1,455,556 (sample: 172 farmers) | 1,721,579 (sample: 182 farmers) | +266,013 (plus 18.3%) |
| Value motorbike or bicycle (TZS) | 243,443 (sample: 97 farmers) | 275,786 (sample: 103 farmers) | +32,343 (plus 13.3%) |
| Education expenditures (TZS) | 132,186 (sample: 90 farmers) | 149,411 (sample: 90 farmers) | +17,225 (plus 13.0%) |
| Health expenditures (TZS) | 65,714 (sample. 44 farmers) | 62,150 (sample: 39 farmers) | −3564 (plus 5.4%) |
| Other assets (TZS) | 647,521 (sample: 71 farmers) | 804,980 (sample: 100 farmers) | +156,970 (plus 2.4%) |

Table 3 shows that not only the use of land and inputs increased between year 1 and 2 but also the weekly consumption as well as monthly farm income increased substantially. Maize production and productivity also increased, although the price of maize was lower in year two, which meant that the turnover per farm increased very little.

We found that many farmers did not invest and those who invested rarely did it in agriculture. In addition to the investments in land already mentioned, in the first year three farmers invested in a deep well, nine in a pump, and three in a tractor, or using tractor services. Some were buying a plow (16 farmers, often oxen-drawn plows), other equipment (6), or tools (32 farmers). Most of the investments went into buildings (31 farmers, in a particular building, or improving their houses), or making 'other' investments (21 farmers, mainly buying cattle or chicken or spending money on the education of their children or for health purposes). In the second year, another three farmers invested in a deep well, six in a pump, and four in a tractor, or using tractor services. Agricultural-related investments were again minimal: buying a plow (3 farmers), other equipment (4), or tools (23). Most investments went again into buildings (36 farmers, in particular houses) or others (33, buying cattle or chicken and family-related expenses, such as health and education of the children). The figures showed there was an ecosystem developing, where farmers can buy these inputs and obtain technical services.

We also identified some problems. Farmers often did not know how much they paid for crop insurance. However, they were generally positive about it. It seems the insurance offers a feeling of security. Indeed the intermediary organizations can reduce the size of the loan in case of a drought. Some farmers were critical of the insurance because no payments were made by the insurance company, although they faced limited rains. Some found that the pay-outs were too low. Many farmers said they wanted another type of support. They said they wanted more transparency concerning the pay-outs from the insurance company and to obtain better prices for their produce.

SCBF should consider consultations with the government, to ensure government support for crop insurance and facilitate the development of an ecosystem to increase the productivity of farmers, of which providing insurance is an important part. This private-sector initiative has helped farmers to run risks and become more entrepreneurial, buying or leasing additional land to increase their income. This was possible because of changes in Tanzania's land policies.

The next table looks at the effects of the introduced innovations on a number of assets of the interviewed farmers between the first and the second year. The value of their assets was quantified. In addition to the land, we looked at houses built or improved, purchase of means of transportation, expenditures on education, expenditures on health, and the value of other assets.

The prices of houses ranged from a few hundred thousand to 15 million TSh. The average value for those 172 farmers owning a house was 1,455,556 TSh. (28 farmers (14%) do not declare that they own a house in the first year.). Similarly, we calculated the average value of motorbikes and/or bicycles, education expenditures, health expenditures, and the value of other assets in the first and the second year. In the second year, only 17 farmers did not declare that they owned a house and for one farmer, we did not have the answer. We came across examples where a good harvest in the first or the second year helped the farmer to build a house. The prices of the houses in the second year ranged from a few hundred thousand shillings to 40 million TSh. However, the average value of a house, for those owning a house, was now 1,721,579 TSh. or substantially higher than in year 1.

Overall, we saw a net increase in most variables, including the value of assets between 2015/2016 and 2016/2017, including the value of houses and motorbikes or bicycle, expenditures on education, and other assets. The results of the survey showed positive effects of the intervention when comparing data for the first and the second year. The project is relevant and had an impact. In addition to indicators of the effects, also the outreach, efficiency, and effectiveness of the interventions were analyzed and positive evidence was found.

We saw a net increase in the value of the assets between 2015/2016 and 2016/2017. For those owning houses, the value increased on average by 266,013 TSh. It should be noted that the order of priority after a good harvest seems to be 1. improve the house, 2. more money for education and health expenditures, and 3. investments in agriculture. In addition, 97 farmers owned bikes or motorbikes (valuing 243,443 TSh. in year 1) and 103 in year two (valuing 275,786 TSh.), showing an increase in the number of farmers having these assets and their average wealth of 32,343 TSh. One hundred and ten farmers declared no expenditure was made for education in 2015/2016 and 2016/2017, but the other 90 farmers spent on average 132,186 TSh. in the first year and 149,411 TSh. in the second year, showing again an increase of 17,225 TSh.

The picture for expenditures on health was slightly different. One hundred and fifty-six farmers did not declare these expenditures, but the other 44 spent on average 65,714 TSh. in year one and 39 farmers 62,150 TSh. in the second year. However, it would be difficult to prove that fewer people spent money on health care and that the lower average expenditures were due to the introduction of crop insurance. The picture for 'other assets' was again quite similar to that of most categories of assets. Seventy-one farmers had other assets and on average, in year 1, worth 647,521 TSh. and in year 2, valuing 804,950 TSh., a substantial increase of 156,970 TSh. per household.

To conclude we found for each category of assets an increase in average spending, while inflation was limited in 2017. In addition, the number of farmers mentioning these assets increased between year one and year two. The exception was the average expenditure on health care, which declined somewhat, just like the number of farmers mentioning health care expenditures for the last year. However, it would be difficult to prove that this is statistically significant and due to the interventions.

## 5. Discussion, Linking the Findings to the Theoretical Framework

The theoretical framework pointed to the importance of land security, external drivers, the ecosystem, and frugal innovations in particular. The land laws and regulations granted customary rights of occupancy equal status to other property rights. It created de facto land ownership in Tanzania. This has made the functioning of a land market possible. However, external drivers were necessary to trigger an emerging land market, which was also the result of migration out of the rural areas (hence, less labor is available, and people may not be able to use all their land) and the frugal innovations described, which were introduced through private-sector initiatives (NGOs and private companies). An ecosystem needs to be in place and consist, in this case, of a combination of private sector actors: a seed company, an insurance company, technology, and input suppliers and NGOs.

First, the land security argument. Kabigi et al. [18] claimed the land reforms were not yet effective. However, at the farmers' level, we found land is being bought and sold or leased and we concluded there is an emerging land market, which came into existence because of a combination of the interventions: the new land policies, the developing ecosystem, and the frugal innovations.

Crop insurance was the frugal innovation serving as an external driver, helping small maize farmers in Tanzania to make a step towards more production. Then the small maize farmers needed more land, and, hence, we noted this emerging land market. The project studied was part of the developing ecosystem and contributed to the training of thousands of farmers in the three regions studied and contributed to the observed functioning of the land market. Not only additional agricultural services were provided through the ecosystem but the project also managed to reach many farmers at minimum cost, combining existing technological options. The findings are in line with the theory of frugal innovation, which emphasizes that nothing new is necessary, but a smart combination of existing technologies can have big effects [2].

The process can be summarized as creating the right policy framework, stimulating the development of an ecosystem, allowing private (commercial and non-commercial) initiatives, and looking for a smart combination of interventions, including existing (frugal) technologies, leading to an emerging land market. The trigger was the combination of frugal innovations, which had positive effects, and brought money into the system. People are noticing that hybrid maize works if you have complementary inputs and insurance. Then they jump on the bandwagon and try to get land or more land, which has contributed to the emergence of a real land market.

## 6. Conclusions

For the success of introducing frugal innovations such as maize crop insurance in Tanzania, the need for a functioning land market has been shown. Crop insurance helps small maize farmers in Tanzania to make a step towards producing more, using more land by buying or leasing it, and using it more intensively. The crop insurance project is successful and has achieved the original objective of reaching 15,000 farmers. The delivery of an all-inclusive package to farmers through a local NGO with regular interaction with the farmers, within an institutional support structure or ecosystem, which can be described as a Triple Helix structure [21], delivered the best results. Crop insurance is in particular useful if it is embedded in an institutional support structure that is non-commercial and close to farmers. It should be provided through a combination of inputs involving local NGOs, which would also make it part of a Climate Smart Agriculture and Water (CSA&W) approach. The insurance concerns the lack of rain. However, drought was not the biggest problem in 2018, but was the destructive effects of caterpillars, not covered in the insurance and difficult to identify with satellites.

There is an unsatisfied demand for crop insurance in Tanzania, from other regions, for other crops and for additional risks (such as damage due to caterpillars). More information and training should be provided to farmers and the insurance process needs to be made more transparent. It is important to make the transition from traditional to hybrid seeds. We have shown the potential of hybrid seed to increase farmers' incomes, to improve food supply, and to contribute to food security in Tanzania.

It is also important to carefully select the intermediary for providing crop insurance. It is better to supply crop insurance as part of a package, which should include hybrid seeds, fertilizers, and eventually, additional inputs such as pesticides and access to water.

The innovation would even be more frugal if mobile phones were always used for the registration of the insurance, to inform the farmers, and for transferring the pay-outs, which is currently not always the case. In general additional information and training should be supplied to farmers and the cost (and benefits) of the insurance should be made more transparent. As mentioned, complaints of the farmers should be taken seriously and should lead to an adaptation of the system to cater to their needs.

Changes in legislation created the conditions for an emerging land market and the people took it up when they were triggered by the introduction of more modern farming methods, while leasing land provides flexibility and allows elderly people or people with too much land, given their available labor and their level of technology, to still reap benefits from their land.

Land lease does not endow the lessee with the full rights to manage the leased land freely, unlike ownership rights. However, appropriate legal regulations may approximate land lease to land ownership in terms of rights. Ensuring that the lessee has an adequate time perspective to work on leased land would encourage him or her to invest in the farm. A chance to renew the lease contract or to have the priority in purchasing leased land would mean that the rights to such land would not be much weaker than their full ownership, and this would contribute to more effective use of land resources by lessees. (This point was made by an anonymous referee of the article.)

The impact of the size of the land was analyzed, showing that the bigger farmers (in terms of land size) benefited relatively more from the opportunity to get more land and inputs and to insure themselves against crop failure.

Weather-index-based crop insurance is a crucial part of the innovation package provided to strengthen the livelihoods of farmers. However, farmers require more support than providing local training activities. They should also be linked to agricultural finance, which is currently missing in the ecosystem. What is also needed next to hybrid seeds and insurance are basic inputs, such as water (irrigation opportunities), more land, improved land management, and knowledge of and access to modern agricultural technologies.

Continuing to modernize agriculture, increase rural incomes and food supply, and, hence, improve food security is recommended. Land markets play an increasingly important role and their functioning should be facilitated. Another recommendation is to develop the link between the farmers and agrarian credit and micro-finance institutions [22]. These institutions should be part of the ecosystem and could institutionalize the loan part.

The transition from traditional maize to hybrid seeds to modernize agriculture increases rural incomes and food supply and is a necessary step in agricultural development. In the first stage, agricultural modernization CSA&W leads to more demand for land. However, if a climate-smart agriculture approach is followed, eventually more production is possible on less land, using modern technology, irrigation, and greenhouses. In the current stage of agricultural development in Tanzania, the developments result in more demand for land. If further technological innovations are introduced, land may become less important, because a few acres can be enough to make a living if greenhouses are used, and input use and disease control are optimized.

The limitations of the research are linked to the limited number of regions and crops studied and the size of the sample. The research was not exclusively focused on land. However, this allowed providing a broader picture of agricultural transformation leading, in the case of appropriate land policies, to an emerging land market.

**Funding:** This research received no external funding.

**Data Availability Statement:** Not applicable.

**Conflicts of Interest:** The authors declare no conflict of interest.

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
