# Peer review of "Crop Insurance, a Frugal Innovation in Tanzania, Helps Small Maize Farmers and Contributes to an Emerging Land Market"

_land, doi:10.3390/land11070954_

Round 1

Reviewer 1 Report

The manuscript takes up an important, current research problem with wide theoretical and practical implications.

The problem of improving the efficiency of work in agriculture is very important and active for many countries. It is not only China that is struggling with it, but also, for example, European Union countries, especially Central and Eastern Europe (including, for example, Poland). The use of the lease to improve unsatisfactory economic indicators is a very good idea, with both scientific and practical justifications.

Land lease does not endow the lessee with the full rights to manage the leased land freely, unlike ownership rights. However, appropriate legal regulations may approximate land lease to land ownership in terms of rights. Ensuring that the lessee has an adequate time perspective to work on leased land would encourage him to invest in his farm (costs of outlays would generate expected profits if a lease contract was long enough). A chance to renew the lease contract or to have the priority in purchasing leased land would mean that the rights to such land would not be much weaker than their full ownership, and this would contribute to a more effective use of land resources by lessees. The area structure of farms can be improved considerably by the transfer of land from economically weaker to stronger farms (and it is not the formal but effective transfer of rights to land that matters here).

The reviewer has no objections or comments to the empirical part and methodology. 

Author Response

Crop insurance, a frugal innovation in Tanzania, helps small maize farmers and contributes to an emerging land market

Land responses Meine Pieter van Dijk (MPVD) in CAPITALS,

VERSION 13-6-2022

Reviewer 1

The manuscript takes up an important, current research problem with wide theoretical and practical implications.

MPVD THANK YOU

The problem of improving the efficiency of work in agriculture is very important and active for many countries. It is not only China that is struggling with it, but also, for example, European Union countries, especially Central and Eastern Europe (including, for example, Poland). The use of the lease to improve unsatisfactory economic indicators is a very good idea, with both scientific and practical justifications.

MPVD I FULLY AGREE

Land lease does not endow the lessee with the full rights to manage the leased land freely, unlike ownership rights. However, appropriate legal regulations may approximate land lease to land ownership in terms of rights. Ensuring that the lessee has an adequate time perspective to work on leased land would encourage him to invest in his farm (costs of outlays would generate expected profits if a lease contract was long enough). A chance to renew the lease contract or to have the priority in purchasing leased land would mean that the rights to such land would not be much weaker than their full ownership, and this would contribute to a more effective use of land resources by lessees. The area structure of farms can be improved considerably by the transfer of land from economically weaker to stronger farms (and it is not the formal but effective transfer of rights to land that matters here).

MPVD THANK YOU FOR MAKING THIS POINT, WHICH I WILL INCLUDE IN THE RECOMMENDATIONS

The reviewer has no objections or comments to the empirical part and methodology. 

MPVD THANK YOU

Reviewer 2 Report

I appreciate your revision and I have no other suggestions.

Author Response

Reviewer 2

I appreciate your revision and I have no other suggestions.

MPVD THANK YOU

Reviewer 3 Report

The paper has been improved after the first round of review. Nonethless the Title can not be accepted in a present form "An emerging land market is helping small maize farmers in Tanzania to benefit from frugal innovations" as it is not scientific but rather a marketing reference. It needs to be changed to one that is more objectified and does not impose specific conclusions. Therefore it could be framed as "An impact of land market on using frugal innovations by small maize farmers in Tanzania" or similar.

Author Response

Reviewer 3

The paper has been improved after the first round of review.

MPVD THANK YOU

Nonethless the Title can not be accepted in a present form "An emerging land market is helping small maize farmers in Tanzania to benefit from frugal innovations" as it is not scientific but rather a marketing reference. It needs to be changed to one that is more objectified and does not impose specific conclusions. Therefore it could be framed as "An impact of land market on using frugal innovations by small maize farmers in Tanzania" or similar.

MPVD I SUGGEST: Crop insurance, a frugal innovation in Tanzania, helps small maize farmers and contributes to an emerging land market

THANK YOU

Reviewer 4 Report

First of all, I think this paper need an additional subsection for section 3, which could focus on explain the relation between land market and frugal innovation diffusion/adoption or discuss the impact of land market on frugal innovation diffusion/adoption.

I am confused about the topic of this manuscript. Does this paper intends to investigate the impact of land market on frugal innovation diffusion or analyze the impact of frugal innovation diffusion on the development of land market, or the interaction between frugal innovation diffusion and the development of land market?

Please give a clear definition for frugal innovation”ï¼›

Line 138: “frugal innovations adoption” could be more accurate than “frugal innovations function”;

I would suggest combine the section 3.3 and section 3.4.

Author Response

Reviewer 4

First of all, I think this paper need an additional subsection for section 3, which could focus on explain the relation between land market and frugal innovation diffusion/adoption or discuss the impact of land market on frugal innovation diffusion/adoption.

MPVD THIS IS MADE CLEAR THROUGH THE NEW TITLE AND IS EXPLAINED IN THE ABSTRACT AND CONCLUSIONS (SEE TRACK CHANGES)

I am confused about the topic of this manuscript. Does this paper intends to investigate the impact of land market on frugal innovation diffusion or analyze the impact of frugal innovation diffusion on the development of land market, or the interaction between frugal innovation diffusion and the development of land market?

MPVD YOU ARE RIGHT THAT IT ANALYZES THE EFFECT OF FRUGAL INNOVATIONS ON THE DEVELOPMENT OF THE LAND MARKET.

MPVD HENCE I SUGGEST: Crop insurance, a frugal innovation in Tanzania, helps small maize farmers and contributes to an emerging land market

Please give a clear definition for “frugal innovation”ï¼›

MPVD DEFINITIONS ARE PROVIDED IN LINE 10, 41, 52, 338-342 AND REFERENCES 2, 4 AND 5

Line 138: “frugal innovations adoption” could be more accurate than “frugal innovations function”;

MPVD I HAVE TAKEN UP YOUR SUGGESTION. THANK YOU

I would suggest combine the section 3.3 and section 3.4.

MPVD FOR ANALYTICAL REASONS I PREFER TO SEPARATE THE ANALYSIS OF THE ECOSYSTEM FROM THE ANALYSIS OF THE IMPACT OF FRUGAL INNOVATIONS

THANKS TO ALL REVIEWERS, I HAVE ADDED AN END NOTE FOR THE SUGGESTIONS MADE BY REVIEWER 1, WHICH I HAVE INCLUDED IN THE CONCLUSIONS.

YOURS

MPVD 14-6-22

Round 2

Reviewer 4 Report

I have no further comments.

This manuscript is a resubmission of an earlier submission. The following is a list of the peer review reports and author responses from that submission.

Round 1

Reviewer 1 Report

In the manuscript submitted for review, a current and important research problem was taken up.

The manuscript has high cognitive and application values. It was prepared with the use of appropriately selected literature on the subject. The methodological assumptions, the obtained research results and conclusions do not raise any doubts of the reviewer. I recommend the manuscript for publication in Land in the version submitted for review.

Reviewer 2 Report

The article has publication potential, but needs to improve the statistical analysis of the results. It is necessary to insert metrics for comparison between groups of farmers and present differences between them.

I suggest inserting an exploratory statistical analysis that demonstrates the portrait of farmers and their socio-economic profile. If possible a comparison with the average of farmers in the country.

Reviewer 3 Report

I thank the author for giving me the opportunity to read his interesting work. I have just a few suggestions to give.
In the conclusions it would be interesting to indicate the limits of the research and possible future developments, as well as the implications for policy makers.
I is also interesting to highlights the contributions of the solution indicated in terms of food security and the permanence of the population in rural areas, avoiding uncontrolled urbanization.

Reviewer 4 Report

My suggestions:

  1. About the title: maybe the opposite.
  2. The title does not correspond to the content of the paper. 
  3. The main subject in the title is the land market. But finally, only a few lines in the text talk about the land market.  
  4. A scientific paper has to have a theoretical background. In this paper, there isn't.  It could be land policies or diffusion of innovation?
  5. What is the land policy in Tanzania and land reforms? It is not clear.
  6. What is an ecosystem? There is an abundant bibliography about it. What is frugal innovation? It is not clear.
  7. lines 51-55: farmers have to choose one package or they can choose more? 
  8. line 85: investors are locals or foreigners? It's important.
  9. 200 farmers were interviewed. What is the total number of farmers? What does the number of 200 farmers represent?
  10. lines 205-219: I think changes are not important.
  11. lines 232, 235, 238: how do they pay? per acre? how match?
  12. Table 3: show changes in %.
  13. Conclusions are not coherent.  

Reviewer 5 Report

From the very beginning – from the title – the paper provides the message that sounds like an advertisement „An emerging land market is helping small maize farmers in Tanzania to benefit from frugal innovations”.

Nonetheless, there is a significant unclearness about the subject of the paper. Does it speak about the land market or about frugal innovations or about crop insurance or about the hybrid seeds sold together with other inputs (called as ecosystem)?

In fact the the paper does not speak primarily about the land market but about crop insurance connected with hybrid seeds and other inputs sale to small farmers. The confirmation of the above statement can be found in conclusions, esp. In lines 440-443 (440 The impact of the size of the land of the farmers was analyzed, showing that there is something like a land market in Tanzania and that the bigger farmers (in terms of land  size) benefit relatively more of the opportunity to get inputs and insure themselves for crop failure); and in the lines 459-460 (The transition from traditional maize to hybrid seeds to modernize agriculture, in- 459 crease rural incomes and food supply is a necessary step in agricultural development.).

The Author described changes in land utilization based on the opinion of 200 farmers (chapter 3.2. Land ownership: an emerging land market?) however not about the land market. We do not know land prices or lease prices, we do not know the size of demand or the size of supply.

The main part of the paper (chapters 3.3 and 3.4) is focusing on the effects of using insurance by registering for a replanting guarantee (RPG by Seedco), or go for a crop insurance for one or more acres of land with or without package of inputs (lines 228-230), with some information about a number of agricultural variables  (table 3) and some indicators of change in household assets (table 4) (lines 282-284). It was however not confirmed if there is any statistical significant relations between analysed variables.

The discussion is limited to two citations used only as illustration not the argument.

In the discussion, the Autor is stating (lines 380-385) „However, at the farmers’ 380 level we find land is being bought and sold or leased and we conclude there is an emerg- 381 ing land market, which came into existence because of a combination of the interventions: 382 the new land policies, the developing ecosystem and the frugal innovations. Crop insur- 383 ance was the frugal innovation, helping small maize farmers in Tanzania to make a step 384 towards more production.”.

Thus, there is a question: what is helping what? Is land market helping farmers to use innovations (as it is in the title) ora are innovations helping the development of land market?.

Only in 7 lines (310-317) the Author is providing information about stated problems by the farmers claiming that this should be solved by the government support to promoted solutions provided by private companies „Complaints of farmers should be taken seriously. SCBF should consider consulta- 318 tions with the government, to ensure government support for crop insurance and facilitate 319 the development of an ecosystem to increase productivity of farmers, of which providing 320 insurance is an important part.”

It is not clear also why the Author is writing „we”.

In conclusion, I think that the main message of the paper is in the lines 265-268 „NGOs 265 or academic institutions may lobby for it and show that there is a business model of the 266 small farmer if he or she can be convinced to use more and different inputs and take a 267 crop insurance”.

Having in mind that the paper has no clear subject of the research and the conclusions are not proven in a scientific way as well as the Author does not support the findings with the scientific discussion I suggest to the Editor reject the paper.